# The UK Soft Drinks Industry Levy and childhood hospital admissions for asthma in England

Nina T. Rogers [1] ✉, Steven Cummins [2], Catrin P. Jones [1], Oliver T. Mytton[3], Chrissy H. Roberts [4], Seif O. Shaheen[5,6], Syed Ahmar Shah [7], Aziz Sheikh [7,8], Martin White [1] & Jean Adams[1]

Sugar sweetened beverage consumption has been suggested as a risk factor for childhood asthma symptoms. We examined whether the UK Soft Drinks Industry Levy (SDIL), announced in March 2016 and implemented in April 2018, was associated with changes in National Health Service hospital admission rates for asthma in children, 22 months post-implementation of SDIL. We conducted interrupted time series analyses (2012-2020) to measure changes in monthly incidence rates of hospital admissions. Sub-analysis was by age-group (5-9,10-14,15-18 years) and neighbourhood deprivation quintiles. Changes were relative to counterfactual scenarios where the SDIL wasn't announced, or implemented. Overall, incidence rates reduced by 20.9% (95% CI: 29.6-12.2). Reductions were similar across age-groups and deprivation quintiles. These findings give support to the idea that implementation of a UK tax intended to reduce childhood obesity may have contributed to a significant unexpected and additional public health benefit in the form of reduced hospital admissions for childhood asthma.

The UK Scientific Advisory Committee on Nutrition (SACN) recommends that consumption of free sugar should be below 5% of total energy intake. In the UK, consumption across all age groups is at least twice this, and three times greater in adolescents aged 11–18 years[1]. Sugar sweetened beverages (SSBs) are a major source of free sugar in the UK diet[2] and high intake has consistently been found to be associated with risk of non-communicable diseases, including obesity, diabetes, and cardiovascular disease[3]. There is also increasing evidence of a link between SSB consumption and incidence of asthma[4], one of the most common diseases in childhood and for which the UK has the highest mortality rates in Europe[5].

A meta-analysis based on one cohort study and 11 cross-sectional studies found an increased odds of asthma prevalence when comparing the highest versus lowest consumers of SSBs[6]. Cross-sectional studies have reported associations[7,8], including dose-response relationships[9], between consumption of SSBs and prevalence of asthma in children. Furthermore, a high percentage of energy from sugars in SSBs was associated with asthma traits in children in the second year of life[10]. A birth cohort study in the US reported that high intake of fructose-containing beverages in early life (~3 years old) was associated with an increased risk of ever having asthma (doctor-diagnosed and including taking asthma

[1]MRC Epidemiology Unit, Institute of Metabolic Science, University of Cambridge School of Clinical Medicine, Cambridge, UK. [2]Population Health Innovation Lab, Department of Public Health, Faculty of Public Health & Policy, London School of Hygiene & Tropical Medicine, London WC1H 9SH, UK. [3]Great Ormond Street Institute of Child Health, 30 Guilford Street, London, UK. [4]Clinical Research Department, London School of Hygiene & Tropical Medicine, London, UK. [5]Wolfson Institute of Population Health, Barts and The London School of Medicine and Dentistry, Queen Mary University of London, London, UK. [6]Allergy and Lung Health Unit, School of Population and Global Health, University of Melbourne, Melbourne, VIC, Australia. [7]Asthma UK Centre for Applied Research, Usher Institute, University of Edinburgh, Edinburgh, UK. [8]Nuffield Department of Primary Care Health Sciences, University of Oxford, Oxford, UK. ✉ e-mail: nina.rogers@mrc-epid.cam.ac.uk

medication or reporting wheezing in last 12 months) by mid-childhood (~8 years old)[11].

While several studies have examined associations between SSB consumption and prevalence or incidence of asthma, we are unaware of any quasi-experimental studies that have investigated the long-itudinal impact of changes in sugar content of drinks on asthma out-comes. In March 2016, the UK government announced that a two-tier Soft Drink Industry Levy (SDIL) on drinks manufacturers, designed to incentivise them to reformulate their drinks, would come into force in April 2018[12], presenting a unique opportunity for a 'natural experiment' to examine potential health impacts of the SDIL. Under the SDIL, manufacturers of soft drinks containing ≥8 g of sugar/100 ml and those with ≥5 to <8 g of sugar/100 ml were subject to a levy of £0.24/litre and £0.18/litre, respectively. Soft drinks containing <5 g/100 ml sugar, 100% fruit juice and milk and milk based drinks and drinks with ≥1.2% alcohol by volume were levy-exempt.

Evidence suggests that the proportion of SDIL-eligible soft drinks available in the UK with >5 g of sugar/100 ml fell from 49% before the announcement (i.e., 2015) to 15% one year following implementation of the SDIL (i.e., 2019)[13]. Household purchasing of sugar from soft drinks also fell by 8 g/household/week, with associated health benefits in terms of reductions in obesity prevalence in some groups of children[14] and carious tooth extractions[15] in children living in England.

Here, we used National Health Service (NHS) hospital episode data from England, and interrupted time series methodology, to compare trends in the observed incidence of childhood hospital admissions for asthma with a modelled counterfactual scenario where the SDIL was not announced or implemented, 22 months after the SDIL came into force (overall, by age-group and by area-based deprivation). Our ana-lyses show that the UK SDIL was associated with a reduction in inci-dence of hospital admissions for asthma in children aged 5–18, and for children living in all areas regardless of deprivation status. While the UK SDIL was originally implemented to improve childhood obesity levels our findings suggest that there may be additional health benefits in the form of reduced incidence of childhood hospitalisations for asthma.

## Results and discussion

The mean incidence rates of hospital admissions for asthma in children during the pre-announcement and post-announcement period are described in Table 1. This reveals large inequalities, with nearly three times as many children from the most deprived areas being admitted to hospital in the pre-announcement period,

equivalent to 26.4/100,000 persons/month (p/m), than in the least deprived areas (9.3/100,000 p/m). Incidence rates for asthma hospital admissions were higher for younger children with those aged 5–9 having around double the rate of children aged 15–18 years.

### Changes in asthma hospital admissions in relation to the SDIL

In children aged 5–18 years there was an overall absolute reduction in hospital admissions for asthma of 4.0 (2.4, 5.7)/100,000 p/m, or a relative reduction of 20.9% (95% CI: 29.6, 12.2), compared to the counterfactual scenario of no SDIL announcement and no imple-mentation (see Table 1 and Fig. 1). Upward trends were observed in overall asthma admissions until a few months after the SDIL announcement when a downward trend was observed (Fig. 1). Dips in admissions appeared to occur in April and August each year (coin-cident with school holidays). Large spikes were observed in early autumn each year, especially September, a time previously reported to be associated with a sharp peak in childhood asthma hospitalisations[16]. Possible explanations include the start of the new school year, when there is a high exposure to respiratory viruses, and allergens in schools (including dust mites), increased stress from starting school and chil-dren getting out of the routine of taking their preventor inhalers over the summer months[16–18].

In each age group, there were also upward trends in incidence of hospital admission rates for asthma from the start of the study-period, but in all cases, compared to the counterfactual, there were significant reductions 22 months following the implementation of the SDIL (Fig. 2). In age-groups 5–9 and 10–14 years, there were relative reductions of 18.6% (95% CI: 30.0, 7.2) and 24.3% (95% CI: 32.1, 16.5), respectively and visualisations suggested a reversal of the upward trend occurring after the SDIL announcement. For adolescents aged 15–18 years, there was a relative reduction of 15.6% (95% CI: 19.7, 11.5) with a flattening out, but not a reversal of the pre-announcement upward trend in hospital admissions.

Hospital admissions for childhood asthma dropped across all deprivation groups. Absolute reductions were 4.8 (7.4, 2.3)/100,000 p/m and 3.4 (4.4, 2.3)/100,000 p/m, in the most and least deprived quintiles, respectively. Comparable relative reductions were 15.5% (95% CI: 23.7, 7.2) and 26.4% (96% CI: 34.6, 18.1), respectively (Fig. 3). Absolute reductions were relatively consistent across the different IMD quintiles, however, some evidence for a trend in higher relative reductions in less deprived areas were observed (Table 1).

**Table 1 | Mean incidence rates (standard deviation) in the pre and post announcement periods of the UK SDIL and change[a] in hospital admissions per 100,000 population per month for asthma, overall and by age groups and IMD quintiles**

| | Mean Incidence of asthma hospital admissions/ 100,000 p/m | | Change in asthma admissions compared to the counterfactual scenario | |
|---|---|---|---|---|
| | Pre-announcement | Post-announcement | Absolute change | Relative change (%) |
| Total population (age 5–18) | 15.8 (4.0) | 16.5 (3.9) | −4.0 (−2.4, −5.7) | −20.9 (−29.6, −12.2) |
| Age-groups | | | | |
| 5–9 | 22.0 (7.1) | 21.3 (6.7) | −4.3 (−6.9, −1.7) | −18.6 (−30.0, −7.2) |
| 10–14 | 14.9 (3.1) | 15.4 (2.8) | −4.7 (−6.2, −3.2) | −24.3 (−32.1, −16.5) |
| 15–18 | 9.0 (2.0) | 11.3 (2.0) | −2.2 (−2.8, −1.6) | −16.3 (−20.9, −11.7) |
| IMD fifths | | | | |
| IMD1: most deprived | 26.4 (7.03) | 28.6 (7.0) | −4.8 (−7.4, −2.25) | −15.5 (−23.7, −7.19) |
| IMD2 | 17.6 (4.4) | 18.7 (4.5) | −4.9 (−6.94, −2.8) | −22.1 (−31.6, −12.6) |
| IMD3 | 13.3 (3.6) | 13.6 (3.3) | −3.8 (−5.47, −2.1) | −23.2 (−33.4, −13.1) |
| IMD4 | 11.6 (2.9) | 11.4 (2.8) | −3.1 (−4.39, −1.7) | −23.2 (−33.3, −13.0) |
| IMD5: least deprived | 9.3 (2.6) | 9.8 (2.4) | −3.4 (−4.41, −2.3) | −26.4 (−34.6, −18.1) |

[a]Change in incidence rate of hospital admissions is compared to a counterfactual scenario of the UK SDIL not being announced or implemented.

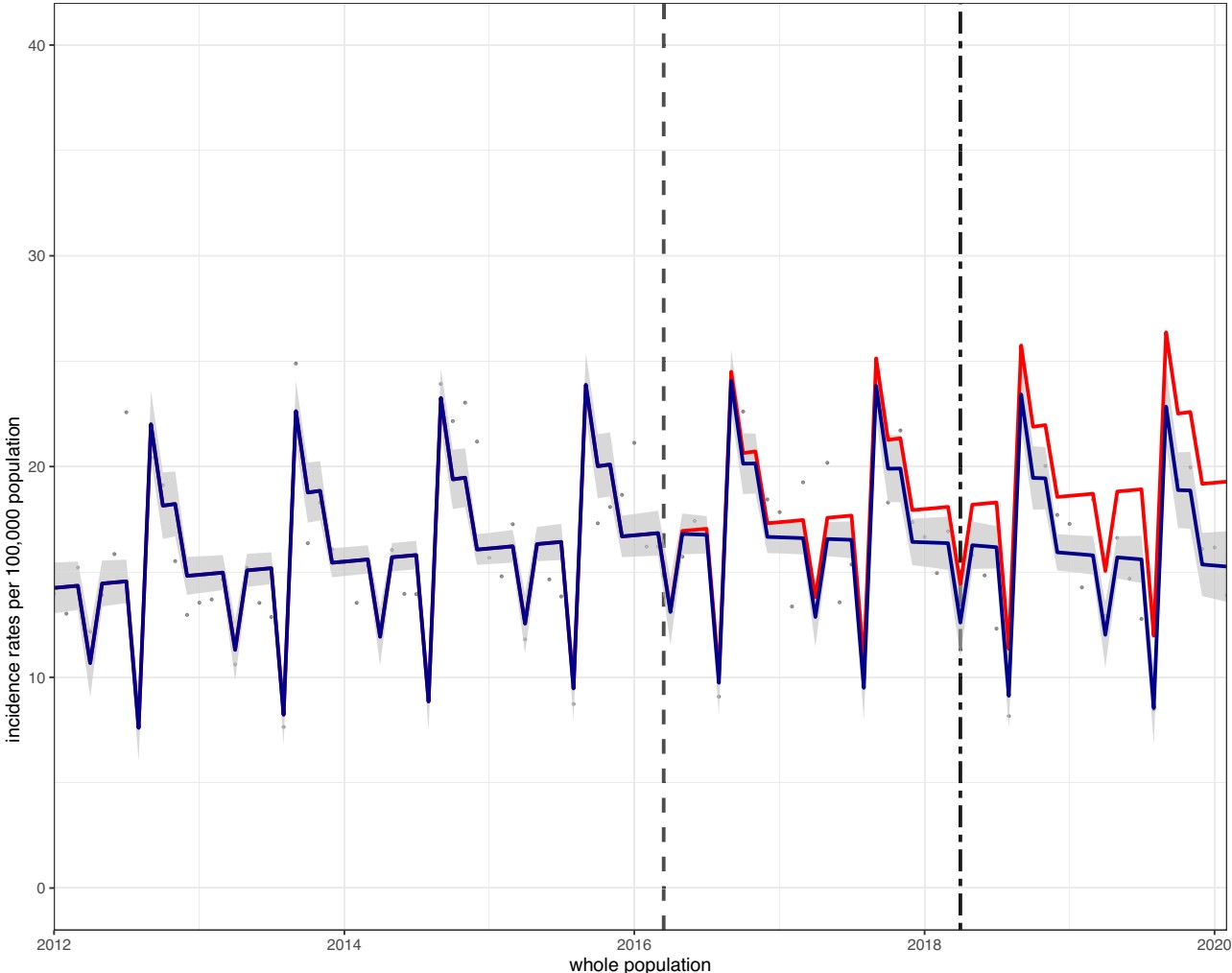

**Fig. 1 | Incidence rates (per 100,000 population per month) of hospital admissions for asthma, in children aged 5–18 between January 2012 and February 2020.** Observed and modelled incidence of hospital admissions for asthma. Dark blue points show observed data and dark blue lines (with grey shadows) shows modelled data (and 95% confidence intervals) of incidence. The red line indicates the counterfactual line based on the pre-SDIL announcement trajectory. The first and second dashed vertical lines indicate the date of the SDIL announcement and implementation, respectively.

## Discussion

In this study we make use of a natural experimental design to provide evidence of a relationship between SSB taxes and asthma outcomes in children. The SDIL was associated with an overall reduction in hospital admissions for asthma in children aged 5–18 years of 21%, 22 months after implementation. Reductions were seen across all areas of deprivation and age groups.

Our findings are consistent with previous studies that suggest an association between SSB consumption and asthma outcomes[7,9–11], but arguably our quasi-experimental design provides stronger evidence for a causal relationship than previous cross-sectional designs. Sugar intake has been linked to the development of asthma, with maternal intake of free sugar found to be a risk factor for asthma and atopy in offspring[19]. Ecological evidence suggests an association between per capita consumption of sugar during the perinatal period and severe asthma in 6–7 year old children[20].

Strengths of our study include using routinely collected hospital admission data on asthma in all children attending an NHS hospital which are therefore not influenced by response bias. We also included a counterfactual scenario in our methodology. One consideration in ITS analysis is the vulnerability to time-varying confounding by other exposures, interventions or treatments that may have been implemented at around the same time as the SDIL, and that are difficult to account for. We are unaware of any such intervention, although acknowledge that in October 2015 a national law that prohibited smoking inside private-vehicles if children were inside came into force[21]. However no association between introduction of this smoke-free vehicle legislation and childhood tobacco smoke exposure or respiratory health has been found, suggesting it is unlikely to be a source of time varying confounding[22]. Outdoor air pollution, particularly ozone, is a major exacerbator of asthma in children; however it is unlikely to have played a role in reducing hospital asthma admissions during the study period because levels have been increasing since 2017[23]. Nitrogen dioxide ($NO_2$), a traffic-related pollutant, has also been suggested to increase incident asthma however levels had been falling for some years before the announcement of the UK SDIL[24]. Lastly, diet quality has been suggested to impact on asthma risk, with the Mediterranean diet, characterised in part by high uptake of fruit and vegetables, reported to be protective against asthma symptoms[25]. Evidence from the UK National Diet and Nutrition Survey suggests that between 2016 and 2019, fruit and vegetable consumption remained stable across the population but did increase by 0.2 portions per day in children aged 11–18[26]. Future work is necessary to examine different dietary components and any potential associations with asthma exacerbation.

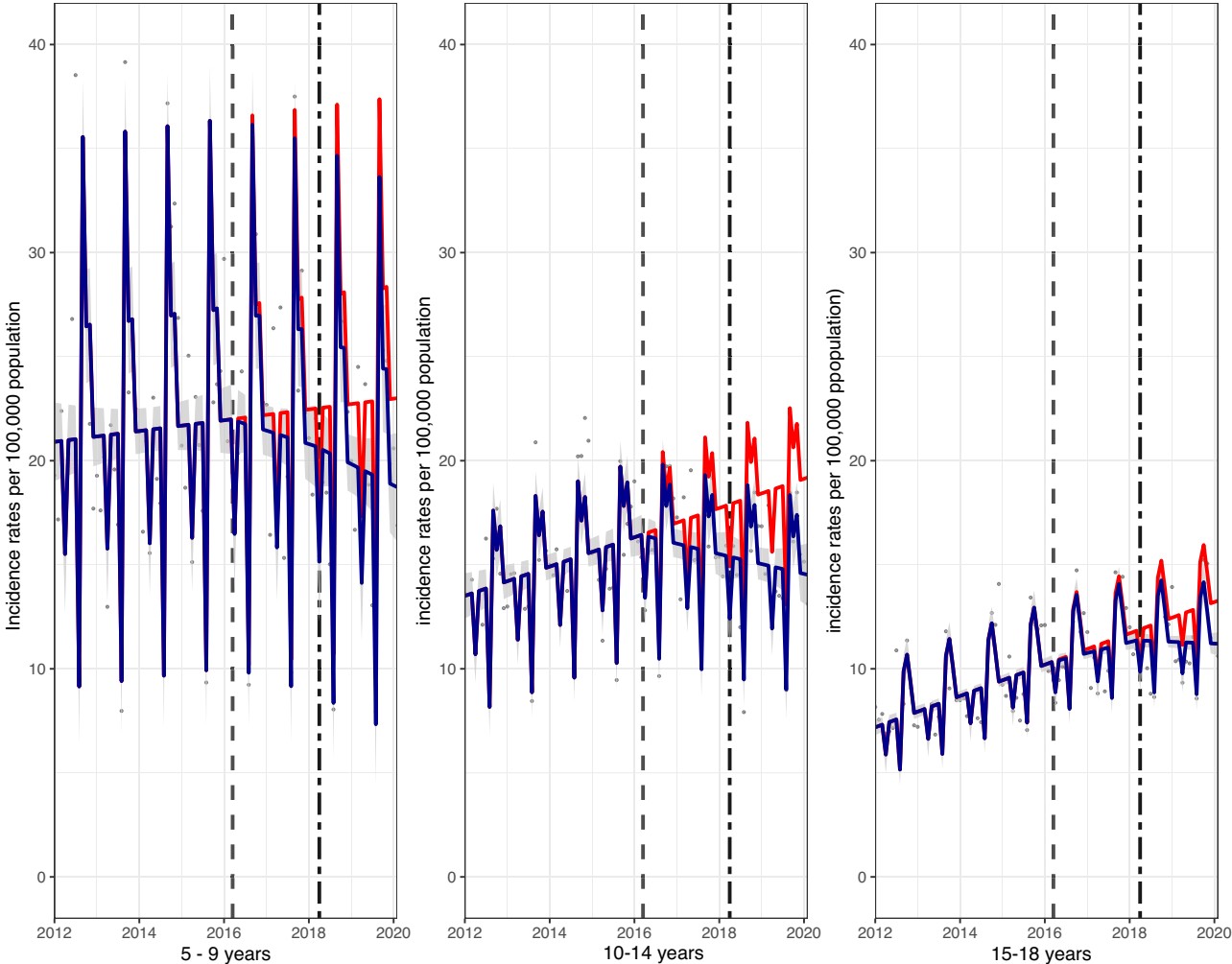

**Fig. 2 | Incidence rates (per 100,000 population per month) of hospital admissions for asthma, in children aged 5–18 between January 2012 and February 2020, by age-group.** Observed and modelled incidence of hospital admissions for asthma. Dark blue points show observed data and dark blue lines (with grey shadows) shows modelled data (and 95% confidence intervals). The red line indicates the counterfactual line based on the observed pre-SDIL announcement trajectory. The first and second dashed vertical lines indicate the time of the SDIL announcement and implementation, respectively.

Our findings link the SDIL to absolute reductions of asthma admissions that are of a similar magnitude across deprivation levels. Lower relative reductions were observed in more deprived areas, indicating that falls in incidence rates of asthma admission were not proportionally as high in children living in more deprived areas. However, this difference is unlikely to reflect changes in sugar consumption. Evidence from previous research suggests that reductions in purchasing of sugar from soft drinks, following the levy, were largest in low vs high income households, perhaps due to higher levels of overall consumption and more scope for lowering levels of sugar[1]. Furthermore, higher price sensitivity (the level of demand in relation to a price change) in those children living in more deprived areas is unlikely to explain the observation because the UK SDIL was not designed to increase prices for the consumer and did not consistently increase the price of eligible drinks[2]. There was a lower relative reduction in hospital admissions in children living in more deprived areas. This may be because the potential benefits of reformulation were partially masked by their cumulative exposures to a plurality of other asthma risk factors. Children living in these communities are at higher risk of exposure to tobacco smoke, pollution, psychosocial stress and sub-optimal management of asthma[3]. A separate study examining asthma admissions in relation to smoke-free vehicle legislation also reported larger relative reductions in asthma admissions in children living in more affluent areas potentially highlighting a similar phenomenon[4].Thus, it seems possible that the UK SDIL alone is unlikely to be sufficient to reverse hospital admissions for asthma at the same proportional levels, unless strategies to reduce other deprivation-related asthma risk factors are implemented alongside SDIL. Compared to the counterfactual scenario, decreases in hospital admissions were observed for all age-groups of children. While the trend in hospital admissions reversed in direction in the younger age-groups, they stabilised in adolescents aged 15–18. Given the lower incidence rates of hospital admissions in this age group one explanation may be the limited potential for hospital admissions to fall much further especially over a short period of time.

Possible explanations for a potential effect of SSB consumption on asthma include obesity (linked to SSB consumption), which is a risk factor for asthma, with factors such as systemic inflammation and nutrition potentially having a moderating effect[27]. On the other hand, most studies have found that that the SSB consumption-asthma link is independent of body mass index or obesity status[7,9–11]. It is also noteworthy that the SDIL was not associated with a significant reduction in the prevalence of obesity in all groups of children[14]. Secondly, preservatives in SSBs have been suggested as a precipitating factor in asthma attacks[28] however studies have not found a link between asthma and diet soft drinks, which also contain the same

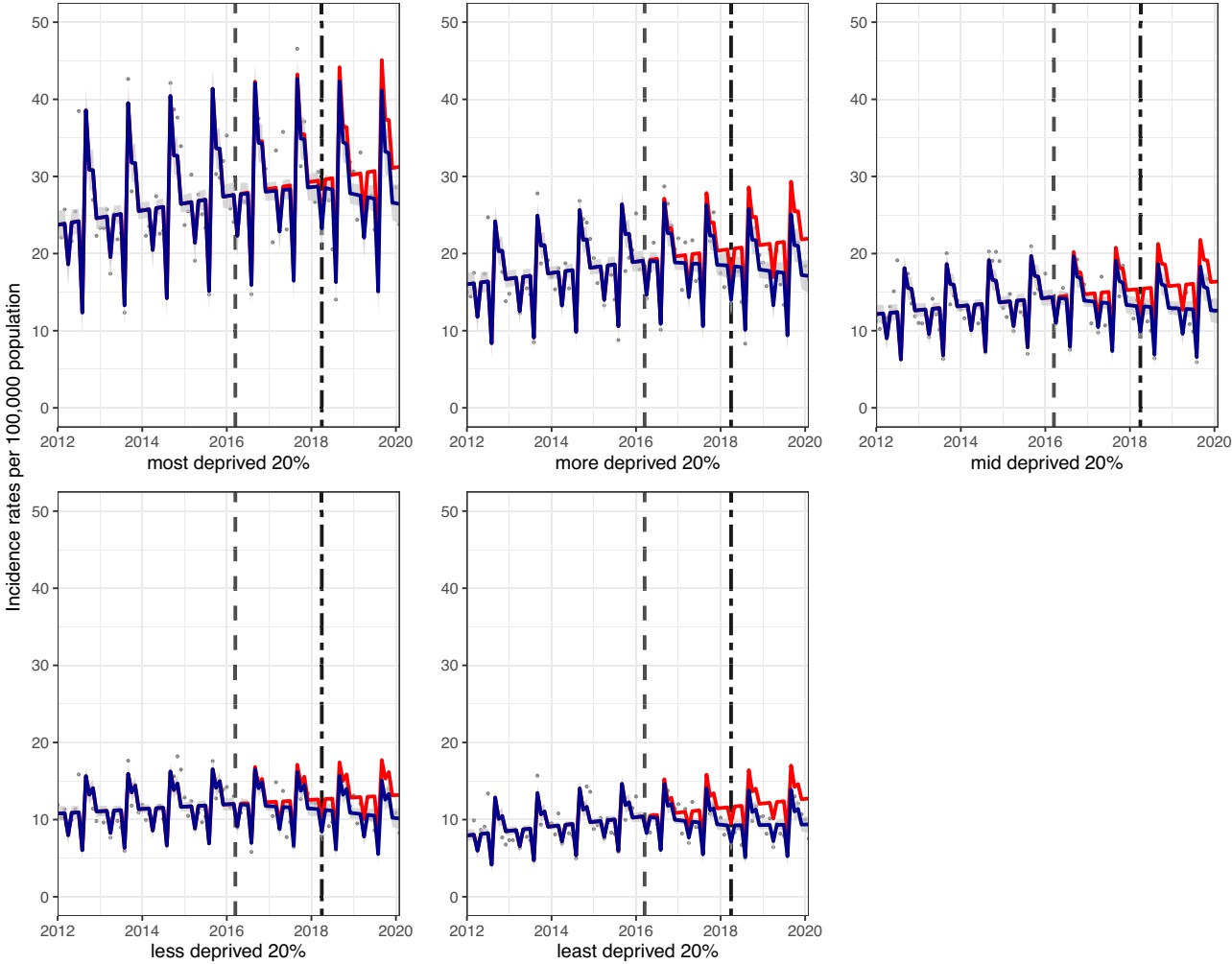

**Fig. 3 | Incidence rates (per 100,000 population per month) of hospital admissions for asthma, in children aged 5–18 between January 2012 and February 2020, by socioeconomic position, measured by Multiple Index of Deprivation quintiles.** Observed and modelled incidence rates of hospital admissions for asthma. Dark blue points show observed data and dark blue lines (with grey shadows) shows modelled data (and 95% confidence intervals). The red line indicates the counterfactual line based on the observed pre-SDIL announcement trajectory. The first and second dashed vertical lines indicate the time of the SDIL announcement and implementation, respectively.

preservatives[7]. Thirdly, high fructose: glucose ratios in SSBs have been implicated in asthma risk with suggestions that reducing consumption may lower asthma risk[4]. Inflammation caused by advanced glycation end products from fructose, may be an important mechanism for the link between fructose consumption and asthma in children. Alternatively, fructose also causes generation of uric acid, and experimental evidence in mice suggests that uric acid promotes Th2 cell-dependent allergic inflammation[29]. This may have important policy implications. For example, 100% fruit juice is a significant contributor of free sugars in the diets of school-age children, contains high levels of free fructose, and has been associated with an increased risk of asthma in children[7,8], but is currently exempt from the SDIL. Other sugar-based drinks (including milk-based drinks, smoothies and powder used to make drinks) are currently not subject to the levy but could be considered as targets for future policies aiming to improve population health. The outcome of this study was restricted to hospital admissions for asthma, and we were unable to capture less severe events in the community. However, a relative reduction of 20% represents a large reduction in severe outcomes. Based on our findings, countries that have not implemented a tax on SSBs may wish to adopt a similar tax to reduce potential cases of hospital admissions for asthma in children.

## Methods

### Data source

National Health Service (NHS) hospital admissions for asthma (International Classification of Diseases; ICD-10 code: J45) in children aged 5–18 y were identified using Hospital Episodes Statistics (HES) data. Data was analysed (1) overall, (2) by age-groups 5–9, 10–14 and 15–18 years and (3) by Index of Multiple Deprivation (IMD) quintile of the Lower Super Output Area (LSOA) of residence[30]. We did not analyse admissions attributed to asthma under five years of age because it is difficult to diagnose asthma at this age when preschool wheeze, a different phenotype, predominates[31]. In HES, age of the patient was calculated from the birth date and episode start date and the 2010 version of the IMD was used to rank LSOAs according to deprivation and assign them into fifths based on the distribution of all LSOAs in England. The study period was from January 2012 (study month 01) to February 2020 (study month 98) and included the time of the SDIL announcement (March 2016; study month 51) and date when the SDIL came into force (April 2018; study month 76). The study was curtailed a month prior to the first national lockdown due to the COVID-19 pandemic to avoid the possibility that admissions after this date may have been related to COVID-related wheezing in children. Data were given

to us in an aggregated and anonymised state and therefore ethical approval was not required for analysis of these data.

**Statistical analysis**

Interrupted time series (ITS) analyses were used to examine the potential impact of the announcement and implementation of the UK SDIL in terms of incidence rates of childhood hospital admissions for asthma. At 22 months post-implementation of the levy (February 2022) the incidence rates of hospital admissions for asthma were compared to a counterfactual scenario where the SDIL was neither announced nor implemented. Data were analysed (1) overall, (2) by age-groups 5–9, 10–14 and 15–18 years and (3) by Index of Multiple Deprivation (IMD) quintile of the Lower Super Output Area (LSOA) of residence[32]. We did not analyse admissions attributed to asthma under five years of age because it is difficult to diagnose asthma at this age when pre-school wheeze, a different phenotype, predominates[27]. The groupwise number of admissions was divided by the respective estimated population size and multiplied by 100,000 to give an incidence rate of hospital admissions per 100,000 population[33]. Time series models were based on generalised least squares (GLS).

Calendar months were tested to determine variation in monthly admissions and GLS models included any month that was associated with significant changes in incidence of hospital admissions. Models were adjusted for September, October, November, April and August, where there were statistically significant changes in incidence rates of hospital admissions.

Counterfactual scenarios were modelled from pre-announcement trend data (study months 01–51). Confidence intervals for calculated differences were estimated from standard errors that were generated using the delta method[34].

Autocorrelation was examined statistically using Durbin–Watson tests and visually using graphical plots of autocorrelation and partial autocorrelation. For individual models we used an autocorrelation-moving average (ARIMA) correlation structure, with the order (p) and moving average (q) parameters specifically selected to minimise the Akaike information criterion (AIC) in individual models.

Statistical analyses were conducted in R version 4.1.0. All code used in the statistical models are included in Supplementary Software 1. Data were acquired through a data sharing agreement with NHS digital for which conditions of use apply. Data requests must be made directly to NHS digital and cannot be granted by the authors.

**Reporting summary**

Further information on research design is available in the Nature Portfolio Reporting Summary linked to this article.

## Data availability

Data can be obtained from a third party and are not publicly available. Data were acquired through a data sharing agreement with NHS digital for which conditions of use apply. Requests for data must be made directly to NHS digital and cannot be granted by the authors. Guidance on how to access the data can be found here https://digital.nhs.uk/services/data-access-request-service-dars

## Code availability

All code used to prepare the data and conduct analysis are provided in Supplementary Software.1

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

## Acknowledgements

N.T.R., M.W. and J.A. were supported by the Medical Research Council (grant reference: MC_UU_00006/7). This project was funded by the NIHR Public Health Research programme (grant references: 16/49/01 and 16/130/01) which were awarded to M.W. The views expressed are those of the authors and not necessarily those of the funders. The funders had no role in study design, data collection and analysis, decision to publish, or preparation of the manuscript.

## Author contributions

N.T.R. and J.A. conceptualised the study. M.W., J.A., S.C. and O.T.M. acquired funding for the study. N.T.R. and J.A. drafted the paper. N.T.R. with help from C.H.R. was responsible for the methodology and conducting the statistical analysis. N.T.R. accepts full responsibility for the work and had access to the data and controlled the decision to publish. All authors were involved in data interpretation and critical reviewing of the manuscript. All authors accept responsibility for the decision to submit the publication.

## Competing interests

The authors declare no competing interests.
