## [Peer Review File · Nature Communications]

Reviewers' Comments:

Reviewer #1:

Remarks to the Author:

The research provides an interesting contribution to the subject of the association between sugar-sweetened beverages and asthma. As a quasi-experimental study, the novelty was the opportunity for a 'natural experiment' to examine the potential impacts of the Soft Drink Industry Levy- SDIL, policy to reduce the consumption of Soft Drink in England, on asthma.

This study investigated the potential impacts of the SDIL on the trends of childhood hospital admissions for asthma (2012-2020), using Interrupted time series (ITS) analyses and comparing it to a counterfactual scenario where the intervention was not announced nor implemented.

The methods are sound to the research question and meet the expected standards in time series studies. However, some issues need to be clarified to strengthen the presentation of the study, especially in the discussion section. My concerns are listed below.

TITLE:

I suggest adding the information: interrupted time series analysis (2012-2020) to the title.

ABSTRACT:

Line 40: ... 22 months post-SDIL (announcement or implementation?). Please clarify.

Line 43: The term "provide evidence" resembles causality. A quasi-experimental design does not allow to making causal inferences. Thus, I suggest replacing it with another term, such as "give support"

Line 45: Please, replace "collateral public health benefit" with "additional public health benefit". In this regard, obesity, the main target of the policy, could be in the path of the association between SSBs and asthma, as a mediator or with an interaction effect.

INTRODUCTION:

Line 63: Please, remove the "recent" (2019) meta-analysis

DISCUSSION:

Lines 138-139

Please, note that a quasi-experimental design does not allow to making causal inferences. Thus, rewrite the sentence as "provides stronger evidence for a causal relationship"

Line 149-150:

The affirmation "It is also noteworthy that the SDIL was not associated with a significant reduction in the prevalence of obesity in all groups of children" differs from the information from lines 82 to 84. Do these different obesity results refer to the same SDIL sugar policies? Please clarify.

Lines 175-176

The lower relative reductions were observed in more deprived areas because incidence rates for asthma hospitalisation are higher in children living in more deprived areas. These results need deep explanation. Was it an unexpected result, since the levy (£0.24/litre and £0.18/litre) may impact the income of the poorest areas? However, is it possible that children living in more deprived areas consumed more SSBs and were less benefitted by health information than least deprived ones?

Please, discuss the results of why the adolescents aged (15-18 years) had not a reversal of the pre-announcement upward trend in hospital admissions. Are they the largest consumers of soft drinks? How about the lowest prevalence in this sub-population at pre-announcement, may it explain these results?

Furthermore, some issues were missing in the discussion. For example, would the short time in trend and some Asthma Phenotypes (atopic asthma; obese, no eosinophilic asthma, etc...) explain the relative change around 20%?

Lines 174-180

The conclusion needs a reformulation. The implications referred just one sub-group discussed. "Our findings indicate that the SDIL alone will not reduce inequalities in asthma hospitalisations

and should be implemented alongside other strategies to reduce inequalities in asthma". Do the authors believe that the relative change of 20% overall population represents an impacting result of policy? And for children? Taxing another sugar-content drink may impact asthma?

Reviewer #2:

Remarks to the Author:

This is an interesting study evaluating the evidence for a policy change related to sugary drinks levies and its impact on asthma hospital admissions.

Since the association between sugar and asthma is still rather speculative, I feel this is an important addition to the evidence base. The evidence of an overall 20% reduction is compelling and the plots illustrate the association well.

My main criticism is that it is difficult to evaluate the structure of the model that was used to evaluate the evidence for change. I suggest that the authors more explicitly state how the policy was evaluated. Was it a binary categorical variable that changed at the time of policy announcement? The plots show three distinct time periods so perhaps a three level categorical variable would be more appropriate (e.g. before, during, after)?

Also, I think it is worth testing for the time period in the study that is associated with greatest evidence for an interval change. For example, this can be implemented with the `tscount::interval_detect()` function in R. It would be interesting to know whether this time coincides with when the intervention was implemented or not. This would strengthen the evidence for the claim that the intervention resulted in a change. At present, the findings are heavily model dependent.

I also note a likely error in line 80-82 of page 3 where it states "Evidence suggests that the proportion of soft drinks available in the UK with <5g of sugar/100 ml fell from 49% before the announcement (i.e., 2015) to 15% one year following implementation of the SDIL (i.e., 2019)". This suggests that lower concentration sugary drink consumption fell in line with the policy, which was the opposite to what I believe was intended.

Otherwise, I feel the manuscript is very well written and presented.

Reviewer #3:

Remarks to the Author:

This study examined whether the UK Soft Drinks Industry Levy (SDIL), announced in March 2016 and implemented in April 2018, was associated with changes in NHS hospital admission rates for asthma in children. Using interrupted time series (ITS) analyses, the authors found that a 20.9% relative reduction in hospital admissions for childhood asthma compared to the counterfactual scenario.

This study provides novel evidence that the implementation of soft drink tax may have reduced hospital admissions for childhood asthma in the UK. This conclusion is important because this study suggests that soft drink tax, which is intended to reduce childhood obesity, may also have beneficial effects on other health outcomes in children.

As with any other ITS analyses, unmeasured or uncontrolled confounding may explain the observed findings. One issue that needs to be discussed is whether there were parallel environmental changes such as reduced air pollution, improved asthma treatment and management, or changes in diet quality that might have contributed to the decline in childhood asthma hospitalization. It is also interesting to see whether asthma hospitalization in adults has experienced the same trend.

Because childhood obesity is associated with both incidence and recurrence of asthma in

childhood, it is useful to understand whether reduction in childhood obesity mediates at least partially the decline in asthma hospitalization during the same time period.

REVIEWER COMMENTS

Reviewer #1 (Remarks to the Author)

The research provides an interesting contribution to the subject of the association between sugar-sweetened beverages and asthma. As a quasi-experimental study, the novelty was the opportunity for a 'natural experiment' to examine the potential impacts of the Soft Drink Industry Levy- SDIL, policy to reduce the consumption of Soft Drink in England, on asthma.

This study investigated the potential impacts of the SDIL on the trends of childhood hospital admissions for asthma (2012-2020), using Interrupted time series (ITS) analyses and comparing it to a counterfactual scenario where the intervention was not announced nor implemented.

The methods are sound to the research question and meet the expected standards in time series studies."

We would like to thank reviewer #1 for conducting a very helpful and thorough review of our paper. We believe that the changes requested by the reviewer have substantially improved the manuscript.

We are also delighted that the reviewer found our research methodology sound and our manuscript novel.

However, some issues need to be clarified to strengthen the presentation of the study, especially in the discussion section. My concerns are listed below.

TITLE:

I suggest adding the information: interrupted time series analysis (2012-2020) to the title.

We thank the reviewer for this suggestion and agree with the addition. The title now reads:

"The UK Soft Drinks Industry Levy and childhood hospital admissions for asthma in England: interrupted time series analysis (2012-2020)"

ABSTRACT:

Line 40: ... 22 months post-SDIL (announcement or implementation?). Please clarify.

We have now clarified that the outcomes we examined were 22 months post implementation of SDIL.

Line 43: The term "provide evidence" resembles causality. A quasi-experimental design does not allow to making causal inferences. Thus, I suggest replacing it with another term, such as "give support".

We thank the reviewer for this suggestion. While interrupted time series analysis does not provide definitive evidence, it is considered to be one of the best designs to establish causality, when RCTs are not possible. However, we have carefully considered the reviewers point and have revised our sentence to read "These findings provide evidence and give support to the idea that the implementation of a tax intended to reduce childhood obesity in the UK may have contributed to a potentially significant unexpected and additional public health benefit in the form of reduced hospital admissions for childhood asthma"

Line 45: Please, replace “collateral public health benefit” with “additional public health benefit”. In this regard, obesity, the main target of the policy, could be in the path of the association between SSBs and asthma, as a mediator or with an interaction effect.

We agree with the reviewer that obesity may lie on the pathway between SSB consumption and asthma and hence we have revised the word “collateral” to “additional”.

INTRODUCTION:

Line 63: Please, remove the “recent” (2019) meta-analysis

We have removed the word “recent”.

DISCUSSION:

Lines 138-139

Please, note that a quasi-experimental design does not allow to making causal inferences. Thus, rewrite the sentence as “provides stronger evidence for a causal relationship”.

As described above, this study does provide evidence through interrupted time series approach, even it is not possible to claim inference. We have edited the sentence accordingly to now read “This is the first study to make use of a natural experimental design to provide evidence of a relationship between SSB taxes and asthma outcomes in children.”.

Line 149-150:

The affirmation “It is also noteworthy that the SDIL was not associated with a significant reduction in the prevalence of obesity in all groups of children” differs from the information from lines 82 to 84. Do these different obesity results refer to the same SDIL sugar policies? Please clarify.

We thank the reviewer for pointing out that these two sentences were not aligned in their message. We have edited the first sentence to make it clear that the UK SDIL was only associated with a reduction in obesity prevalence in some groups of children. The sentence now reads: “Household purchasing of sugar from soft drinks also fell by 8g/household/week, with associated health benefits in terms of reductions in obesity prevalence in some groups of children”.

Lines 175-176

The lower relative reductions were observed in more deprived areas because incidence rates for asthma hospitalisation are higher in children living in more deprived areas. These results need deep explanation. Was it an unexpected result, since the levy (£0.24/litre and £0.18/litre) may impact the income of the poorest areas? However, is it possible that children living in more deprived areas consumed more SSBs and were less benefitted by health information than least deprived ones?

We thank the reviewer for making this helpful suggestion to clarify our results around childhood deprivation. It is true that while absolute reductions were similar across deprivation groups, these decreases were not *proportionally* similar. Those in the more deprived groups experienced smaller relative reductions. A similar result has also been found in studies which looked at the relationship between asthma and “smoke-free car” legislation in the context of a possible societal widening of inequalities¹. We have included more information and references to improve our discussion and have added the following paragraph to the discussion:

“Our findings link the SDIL to absolute reductions of asthma admissions that are of a similar magnitude across deprivation levels. Lower relative reductions were observed in more deprived areas, indicating that falls in incidence rates of asthma admission were not proportionally as high in children living in more deprived areas. However, this difference is unlikely to reflect changes in sugar consumption. Evidence from previous research suggests that reductions in purchasing of sugar from soft drinks, following the levy, were largest in low vs high income households, perhaps due to higher levels of overall consumption and more scope for lowering levels of sugar². Furthermore, higher price sensitivity (the level of demand in relation to a price change) in those children living in more deprived areas is unlikely to explain the observation because the UK SDIL was not designed to increase prices for the consumer and did not consistently increase the price of eligible drinks³. There was a lower relative reduction in hospital admissions in children living in more deprived areas. This may be because the potential benefits of reformulation were partially masked by their cumulative exposures to a plurality of other asthma risk factors. Children living in these communities are at higher risk of exposure to tobacco smoke, pollution, psychosocial stress and sub-optimal management of asthma⁴. A separate study examining asthma admissions in relation to smoke-free vehicle legislation also reported larger relative reductions in asthma admissions in children living in more affluent areas potentially highlighting a similar phenomenon⁵. Thus, it seems possible that the UK SDIL alone is unlikely to be sufficient to reverse hospital admissions for asthma at the same proportional levels, unless strategies to reduce other deprivation-related asthma risk factors are implemented alongside SDIL.

Please, discuss the results of why the adolescents aged (15-18 years) had not a reversal of the pre-announcement upward trend in hospital admissions. Are they the largest consumers of soft drinks? How about the lowest prevalence in this sub-population at pre-announcement, may it explain these results? Furthermore, some issues were missing in the discussion. For example, would the short time in trend and some Asthma Phenotypes (atopic asthma; obese, no eosinophilic asthma, etc...) explain the relative change around 20%?

We have now added some more information regarding the change in trends of hospital admission. The following paragraph has now been included:

“Compared to the counterfactual scenario, decreases in hospital admissions were observed for all age-groups of children. While the trend in hospital admissions reversed in direction in the younger age-groups, they stabilised in adolescents aged 15-18. Given the lower incidence rates of hospital admissions in this age group one explanation may be the limited potential for hospital admissions to fall much further especially over a short period of time.”

We have discussed different possible mechanisms of action including mediation through obesity and inflammation in our discussion. This includes the following details “Possible explanations for a potential effect of SSB consumption on asthma include obesity (linked to SSB consumption), which is a risk factor for asthma, with factors such as systemic inflammation and nutrition potentially having a moderating effect⁶. On the other hand, most studies have found that that the SSB consumption-asthma link is independent of body mass index or obesity status⁷⁻¹⁰. It is also noteworthy that the SDIL was not associated with a significant reduction in the prevalence of obesity in all groups of children¹¹. Secondly, preservatives in SSBs have been suggested as a precipitating factor in asthma attacks¹² however studies have not found a link between asthma and diet soft drinks, which also contain the same preservatives⁹. Thirdly, high fructose: glucose ratios in SSBs have been implicated in asthma risk with suggestions that reducing consumption may lower asthma risk¹³. Inflammation caused by advanced glycation end products from fructose, may be an important mechanism for the link between fructose consumption and

asthma in children. Alternatively, fructose also causes generation of uric acid, and experimental evidence in mice suggests that uric acid promotes Th2 cell-dependent allergic inflammation¹⁴”

Lines 174-180

The conclusion needs a reformulation. The implications referred just one sub-group discussed. “Our findings indicate that the SDIL alone will not reduce inequalities in asthma hospitalisations and should be implemented alongside other strategies to reduce inequalities in asthma”. Do the authors believe that the relative change of 20% overall population represents an impacting result of policy? And for children? Taxing another sugar-content drink may impact asthma?

We have reorganised the discussion and added some text which discusses potential mechanisms in a more logical way. This includes a brief discussion of the possible involvement of fruit juices, which are currently exempt from the UK SDIL. We have also discussed the possibility of including other drinks within the levy and have added the following sentence:

“Other sugar-based drinks (including milk-based drinks, smoothies and powder used to make drinks) are currently not subject to the levy but could be considered as targets for future policies aiming to improve population health. The outcome of this study was restricted to hospital admissions for asthma, and we were unable to capture less severe events in the community. However, a relative reduction of 20% represents a large reduction in severe outcomes. Based on our findings, countries that have not implemented a tax on SSBs may wish to adopt a similar tax to reduce potential cases of hospital admissions for asthma in children.

Reviewer #2 (Remarks to the Author):

This is an interesting study evaluating the evidence for a policy change related to sugary drinks levies and its impact on asthma hospital admissions.

Since the association between sugar and asthma is still rather speculative, I feel this is an important addition to the evidence base. The evidence of an overall 20% reduction is compelling and the plots illustrate the association well.

We thank reviewer 2 for their very helpful comments and are very pleased they thought the study was an important addition to the current evidence base. We hope that the resulting revisions have improved the clarity and quality of the manuscript.

My main criticism is that it is difficult to evaluate the structure of the model that us used to evaluate the evidence for change. I suggest that the authors more explicitly state how the policy was evaluated. Was it a binary categorical variable that changed at the time of policy announcement? The plots show three distinct time periods so perhaps a three level categorical variable would be more appropriate (e.g. before, during, after)? Also, I think it is worth testing for the time period in the study that is associated with greatest evidence for an interval change. For example, this can be implemented with the `tscount::interval_detect()` function in R. It would be interesting to know whether this time co-incides with when the intervention was implemented or not. This would strengthen the evidence for the claim that the intervention resulted in a change. At present, the findings are heavily model dependent.

We thank the reviewer for making helpful suggestions around how we can triangulate our findings that the intervention was related to hospital admissions. We have now supplied our code as supplementary files, which should provide all the detail needed for any user to reproduce our findings. We have clarified the methodology in the statistical analysis subsection of the methods, which now reads as follows:

“Interrupted time series (ITS) analyses were used to examine the potential impact of the announcement and implementation of the UK SDIL in terms of incidence rates of childhood hospital admissions for asthma. At 22 months post-implementation of the levy (February 2022), the incidence rates of hospital admissions for asthma were compared to a counterfactual scenario where the SDIL was neither announced nor implemented. Data were analysed (1) overall, (2) by age-groups 5-9, 10-14 and 15-18 years and (3) by Index of Multiple Deprivation (IMD) quintile of the Lower Super Output Area (LSOA) of residence¹⁵. We did not analyse admissions attributed to asthma under five years of age because it is difficult to diagnose asthma at this age when preschool wheeze, a different phenotype, predominates¹⁶.”

We also appreciate the reviewer's suggestion that we might want to use the functions of the `tscount` package to infer likely points at which change occurred in the admissions during the time-series (i.e. by estimating the tau statistic). There's a robust logic underpinning this, but several issues prevented us from pursuing this approach. The method implemented by the `tscount::interv_detect` function does not lend itself to working with proportions (i.e. the incidence rates we use as the primary outcome in our analysis) as opposed to absolute numbers (i.e. counts) and fails when presented with non-integer data. To use this approach, we would have had to transform the data, which we expect would have then had impacts on the relative statistical relationships between observations because of aberrant scaling.

Furthermore, the `tscount::interv_detect` is susceptible to producing hard-to-interpret results when there is the potential for multiple change-points (as seen in the complex SDIL

intervention). It tends to create local maxima (values of tau) which may not have the highest values of tau overall (i.e. they aren't the absolute maximum for the entire series), but which might still indicate the start of a process of more complex, multi-factorial, multi-modal and/or accelerating temporal change.

The careful specifications we have made to our ARIMA model (and the robust process of model refinement via analysis of delta AIC) accounts for a complex model, whilst correcting for both moving average and autoregressive tendency in the data. Specifying an equally robust *glm* based approach would require some rather complex data transformation and would not be (in our opinion) as robust as what we've already done.

We accept that there could be a question about where change happens in the admissions curves that we present, although simply 'eyeballing' them makes a strong case that the changes initiated at announcement and built with time. For example, *Scarborough et al.(2020)* has shown how reformulation was a gradual process that initiated some months after announcement, but which came closer to completion prior to the commencement of the legislation.³

We believe that the key date in this study is the announcement, which represents a real-world calendar event that happened on a precise date. This was the start of the true "intervention" and whilst we have been careful to avoid making causal claims about the relationship between the intervention and the estimated changes, it is sensible that this date (announcement) should have been used as the point in time from which we extrapolated the counterfactual models. Selecting a time point such as the date of implementation (2018) would not have been sensible given that it is unfeasible that industry would not use the transition period to implement the requirements of the legislation. There is also sound evidence that manufacturers were quick to begin reformulating drinks to contain less sugar several months after the announcement³

I also note a likely error in line 80-82 of page 3 where it states "Evidence suggests that the proportion of soft drinks available in the UK with <5g of sugar/100 ml fell from 49% before the announcement (i.e., 2015) to 15% one year following implementation of the SDIL (i.e., 2019)". This suggests that lower concentration sugary drink consumption fell in line with the policy, which was the opposite to what I believe was intended.

We thank the reviewer for noticing this error. We have now corrected the sentence to read ">5g of sugar/100 ml fell from 49% before the announcement...."

Otherwise, I feel the manuscript is very well written and presented.

Once again, we thank reviewer 2 for their helpful comments and are delighted that they thought the manuscript was well written and presented.

Reviewer #3 (Remarks to the Author):

This study examined whether the UK Soft Drinks Industry Levy (SDIL), announced in March 2016 and implemented in April 2018, was associated with changes in NHS hospital admission rates for asthma in children. Using interrupted time series (ITS) analyses, the authors found that a 20.9% relative reduction in hospital admissions for childhood asthma compared to the counterfactual scenario. This study provides novel evidence that the implementation of soft drink tax may have reduced hospital admissions for childhood asthma in the UK. This conclusion is important because this study suggests that soft drink tax, which is intended to reduce childhood obesity, may also have beneficial effects on other health outcomes in children.

We thank reviewer 3 for pointing out the importance and novelty of our study.

As with any other ITS analyses, unmeasured or uncontrolled confounding may explain the observed findings. One issue that needs to be discussed is whether there were parallel environmental changes such as reduced air pollution, improved asthma treatment and management, or changes in diet quality that might have contributed to the decline in childhood asthma hospitalization. It is also interesting to see whether asthma hospitalization in adults has experienced the same trend.

We appreciate the reviewer's point about adding more detail of other events or exposures that may have influenced hospital admissions for asthma. In addition to the details we have given regarding the smoke-free vehicle legislation, we have now added the following text along with relevant references

“Outdoor air pollution, particularly ozone, is a major exacerbator of asthma in children; however it is unlikely to have played a role in reducing hospital asthma admissions during the study period because levels have been increasing since 2017¹⁷. Nitrogen dioxide (NO₂), a traffic-related pollutant, has also been suggested to increase incident asthma however levels had been falling for some years before the announcement of the UK SDIL. Lastly, diet quality has been suggested to impact on asthma risk, with the Mediterranean diet, characterised in part by high uptake of fruit and vegetables, reported to be protective against asthma symptoms¹⁸. Evidence from the UK National Diet and Nutrition Survey suggests that between 2016 and 2019, fruit and vegetable consumption remained stable across the population but did increase by 0.2 portions per day in children aged 11-18¹⁹. Future work is necessary to examine different dietary components and any potential associations with asthma exacerbation.”

Because childhood obesity is associated with both incidence and recurrence of asthma in childhood, it is useful to understand whether reduction in childhood obesity mediates at least partially the decline in asthma hospitalization during the same time period.

We thank the reviewer for their comments on the importance of exploring whether obesity is an important explanatory factor in the relationship between SSB consumption and asthma. Although we are unable to explore this using the available data we have we have discussed this topic area within our discussion on the possible mechanism of action

References (numbers within this response may differ from the manuscript reference numbers)

- 1 Mackay DF, Turner SW, Semple SE, Dick S, Pell JP. Associations between smoke-free vehicle legislation and childhood admissions to hospital for asthma in Scotland: an interrupted time-series analysis of whole-population data. *Lancet Public Heal* 2021; **6**: e579–86.
- 2 Rogers N, Cummins S, Pell D, *et al*. Changes in household purchasing of soft drinks following the UK Soft Drinks Industry Levy by household income and composition: controlled

- interrupted time series analysis, March 2014 to November 2019. 2023.
DOI:<https://www.medrxiv.org/content/10.1101/2023.11.27.23299070v2.full.pdf>.
- 3 Scarborough P, Adhikari V, Harrington RA, *et al*. Impact of the announcement and implementation of the UK Soft Drinks Industry Levy on sugar content, price, product size and number of available soft drinks in the UK, 2015-19: A controlled interrupted time series analysis. *PLOS Med* 2020; **17**: e1003025.
 - 4 Kopel L, Phipantanakul W, Gaffin J. Social disadvantage and asthma control in children. *Paediatr Respir Rev* 2014; **15**: 256–63.
 - 5 Turner S, Mackay D, Dick S, Semple S, Pell JP. Associations between a smoke-free homes intervention and childhood admissions to hospital in Scotland: an interrupted time-series analysis of whole-population data. *Lancet Public Heal* 2020; **5**: e493–500.
 - 6 Peters U, Dixon AE, Forno E. Obesity and asthma. *J Allergy Clin Immunol* 2018; **141**: 1169–79.
 - 7 Xie L, Atem F, Gelfand A, Delclos G, Messiah SE. Association between asthma and sugar-sweetened beverage consumption in the United States pediatric population. *J Asthma* 2022; **59**: 926–33.
 - 8 Wright LS, Rifas-Shiman SL, Oken E, Litonjua AA, Gold DR. Prenatal and early life fructose, fructose-containing beverages, and midchildhood asthma. *Ann Am Thorac Soc* 2018; **15**: 217–24.
 - 9 Berentzen NE, Van Stokkom VL, Gehring U, *et al*. Associations of sugar-containing beverages with asthma prevalence in 11-year-old children: The PIAMA birth cohort. *Eur J Clin Nutr* 2015; **69**: 303–8.
 - 10 Padilha LL, Vianna EO, Vale ATM, Nascimento JXPT, da Silva AAM, Ribeiro CCC. Pathways in the association between sugar sweetened beverages and child asthma traits in the 2nd year of life: Findings from the BRISA cohort. *Pediatr Allergy Immunol* 2020; **31**: 480–8.
 - 11 Rogers NT, Cummins S, Forde H, *et al*. Associations between trajectories of obesity prevalence in English primary school children and the UK soft drinks industry levy: An interrupted time series analysis of surveillance data. *PLOS Med* 2023; **20**: e1004160.
 - 12 Steinman HA, Weinberg EG. The effects of soft-drink preservatives on asthmatic children. *South African Med J* 1986; **70**: 404–6.
 - 13 DeChristopher LR, Tucker KL. Excess free fructose, high-fructose corn syrup and adult asthma: The Framingham Offspring Cohort. *Br J Nutr* 2018; **119**: 1157–67.
 - 14 Kool M, Willart MAM, van Nimwegen M, *et al*. An Unexpected Role for Uric Acid as an Inducer of T Helper 2 Cell Immunity to Inhaled Antigens and Inflammatory Mediator of Allergic Asthma. *Immunity* 2011; **34**: 527–40.
 - 15 Oxford Consultants for Social Inclusion. Why the Indices of Deprivation are Still Important in the Open Data Era. 2011.
 - 16 Yang CL, Gaffin JM, Radhakrishnan D. Question 3: Can we diagnose asthma in children under the age of 5 years? DOI:10.1016/j.prrv.2018.10.003.
 - 17 Department for Environment Food and Rural Affairs. National Statistics Air quality statistics in the UK, 1987 to 2022 - Ozone (O3). 2023.
 - 18 Chatzi L, Apostolaki G, Bibakis I, *et al*. Protective effect of fruits, vegetables and the Mediterranean diet on asthma and allergies among children in Crete. *Thorax* 2007; **62**: 677–83.
 - 19 Public Health England. NDNS: results from years 9 to 11 (combined) – statistical summary. 2020.

Reviewers' Comments:

Reviewer #1:

Remarks to the Author:

Thank you to the authors for addressing all concerns about the first version. Now, the conclusion brings recommendations and implications for the results. I have no remaining concerns or comments.

Reviewer #2:

Remarks to the Author:

I am satisfied with the responses to my comments. I recommend the manuscript be published.

Reviewer #3:

Remarks to the Author:

The authors did an excellent job in responding to my earlier comments. I have no additional concerns.